

# Sea waves impact on turbulent heat fluxes in the Barents Sea according to numerical modeling

Stanislav Myslenkov[1,2,3] Anna Shestakova[4], Dmitry Chechin[4,5]

[1]Lomonosov Moscow State University, 119991, Moscow, Russia

[2]Shirshov Institute of Oceanology RAS, 117997, Moscow, Russia

[3]Hydrometeorological Research Centre of the Russian Federation, 123242, Moscow, Russia

[4]A.M.Obukhov Institute of Atmospheric Physics RAS, 119017, Moscow, Russia

[5]Moscow Institute of Physics and Technology, 119017, Moscow, Russia

*Correspondence to*: Stanislav Myslenkov (stasocean@gmail.com)

**Abstract.** This paper investigates the impact of sea waves on turbulent heat fluxes in the Barents Sea. The COARE algorithm, meteorological data from reanalysis and wave data from the WW3 wave model results were used. The turbulent heat fluxes were calculated using the modified Charnock parameterization for the roughness length and several parameterizations, which explicitly account for the sea waves parameters. A catalog of storm wave events and a catalog of extreme cold-air outbreaks over the Barents Sea were created and used to calculate heat fluxes during extreme events.

The important role of cold-air outbreaks in the energy exchange of the Barents Sea and the atmosphere is demonstrated. A high correlation was found between the number of cold-air outbreaks days and turbulent fluxes of sensible and latent heat, as well as with the net flux of long-wave radiation averaged over the ice-free surface of the Barents Sea during a cold season.

The differences in the long-term mean values of heat fluxes calculated using different parameterizations for the roughness length are small and are on average 1-3% of the flux magnitude. Parameterizations of Taylor and Yelland and Oost et al. on average lead to an increase of the magnitude of the fluxes, and the parameterization of Drennan et al. leads to a decrease of the magnitude of the fluxes over the entire sea compared to the Charnock parameterization.

The magnitude of heat fluxes and their differences during the storm wave events exceed the mean values by a factor of 2. However, the effect of explicit accounting for the wave parameters is, on average, small and multidirectional, depending on the used parameterization for the roughness length. In the climatic aspect, it can be argued that the explicit accounting for sea waves in the calculations of heat fluxes can be neglected.

However, during the simultaneously observed storm waves and cold-air outbreaks, the sensitivity of the calculated values of fluxes to the used parameterizations increase along with the turbulent heat transfer increase. In some extreme cases, during storms and cold-air outbreaks, the difference reaches 700 W m$^{-2}$.

Keywords: Barents Sea; turbulent heat flux; Charnock parameter; COARE; wind wave hindcast; cold-air outbreaks





## 1. Introduction

Atlantic water undergoes a significant transformation in the Barents Sea where its characteristics, such as temperature, salinity and density, change. New water masses are formed which contain different volumes of the original Atlantic water (Ivanov and Timokhov, 2019). A significant part of the heat content of Atlantic water is spent on melting ice and heating the atmosphere influencing the climatic characteristics of the region (Rahmstorf and Ganopolski, 1999). To a large extent, the heat exchange between the Barents Sea and the atmosphere is carried out by the turbulent heat flux. The Barents Sea is known to be one of the most efficient heat sinks from the ocean to the atmosphere (Simonsen and Haugan, 1996). On average, turbulent heat transfer in the Barents Sea is about 30 W/m2, according to modeling data (Arthun and Schrum 2010). However, even rough reanalysis data show that in energy active zones near the ice edge, fluxes can reach 500 W/m2 (Hakkinen and Cavalieri 1989).The latter depends on the surface roughness, which is associated with the wind wave parameters. Thus, adequate representation of surface roughness is crucial for correct estimates of the surface heat flux.

The modern models of the atmosphere and ocean commonly use the Charnock formula (Charnock, 1955) as a parameterization of the aerodynamic roughness length over the water. The Charnock relationship represents a quadratic dependence of the roughness length on the friction velocity. The Charnock parameter as constant, which represents the proportionality coefficient between the roughness length and the square of friction velocity, used in the most frequently models and reanalyses (for example, in NCEP/NCAR, NCEP/CFSR, MERRA reanalyses). However, numerous studies of roughness behavior in different conditions according to observational data (e.g. Oost et al. 2002, Mahrt et al. 2003) showed that the Charnock parameter (coefficient) is not constant, especially in conditions of high wind speed and high waves. The Charnock formula is applicable when the wave state is in equilibrium with wind forcing, and does not take into account the age of the waves and such effects as wave breaking and spray formation.

Thereby, several parametrizations were proposed that explicitly or implicitly take into account the influence of such wave parameters as wave height, wave length and period on the sea surface roughness.

In the most simple modification of the Charnock formulation the Charnock parameter is set as a piecewise constant or a linear function of wind speed in order to fit the observations. In other parametrizations, the Charnock parameter explicitly depends on the wind wave parameters, usually the wave steepness (Taylor and Yelland 2001) and wave the age (Jones and Toba 2001, Oost et al. 2002, Drennan et al. 2003). More complex parameterizations are based on the relation between the roughness length and the wave momentum flux (Janssen 1991) and are typically used in coupled wave-atmosphere models, including ECMWF operational analysis and reanalyses (ECMWF 2007). Intercomparisons of different roughness parametrizations, including Taylor and Yelland (2001), Oost et al. (2002) and Drennan et al. (2003) parametrizations, did not reveal the best of them (Pan et al. 2008, Charles and Hemer 2013, Shimura et al. 2017, Kim et al. 2018, Prakash et al. 2019). Some studies have shown that Oost et al. parametrization overestimates the roughness of the sea surface in comparison with other schemes (Pan et al. 2008, Kim et al. 2018), and Drennan et al. parametrization usually gives a lower roughness (Charles and Hemer 2013).

The choice of roughness length parameterization affects primarily the momentum flux and turbulent heat transfer. The sensible and latent heat fluxes are calculated using also the roughness length for temperature and specific humidity, respectively. The ratio of the roughness lengths for scalars and momentum is typically parameterized as function of the Reynolds roughness number (Brutsaert 1982, Zilitinkevich et al. 2001, Renfrew et al. 2002, Brunke et al . 2011).

The turbulent heat transfer is parameterized using bulk formulae in most reanalyses, which differ in the choice of the parameterization for the roughness length for temperature and humidity, parameterization of the Charnock parameter, and of the universal functions describing the dependence of the transfer coefficients on the





surface layer stratification (Renfrew et al. 2002, Brunke et al. 2011). A list of the parameterizations used in the main
reanalyses is given in the Appendix by Brunke et al. (2011).
The use of certain parameterization can significantly affect the value of the calculated heat and momentum
fluxes. For instance, the difference in the total turbulent heat flux between the two most commonly used algorithms,
NCAR (Large and Yeager, 2009) and COARE (Coupled Ocean Atmosphere Response Experiment) (Fairall et al.
1996), is 13 W/m$^2$ on average throughout the globe and reaches 15-20% of the flux magnitude in mid-latitudes and
subpolar regions (Brodeau et al. 2017). Typical values of the average difference of turbulent fluxes produced by
different algorithms and the observational data amount to 5-15 W/m$^2$. Unambiguously "the best set of
parameterizations" of the roughness length and universal functions for calculating heat and momentum fluxes does
not exist (Brunke et al. 2011;, Charles and Hemer 2013). Nevertheless, the widely used COARE algorithm (Fairall et
al. 1996, Fairall et al. 2003), which is also embedded in satellite flux calculation algorithms, is considered the most
reliable for calculating turbulent fluxes. Satellite products such as J-OFURO, HOAPS, and OAFlux (joint satellite and
simulation product), use algorithms very similar to COARE (Brunke et al. 2011, Yu et al. 2011). The COARE
algorithm offers a choice of Taylor and Yelland (2001) and Oost et al. (2002) roughness length parameterizations,
which explicitly take into account the wind wave parameters.
Roughness length dependency on wind wave parameters is expected to have regional differences depending
on the local features of the wave regime. According to studies of the wave climate of the Barents Sea (Wind and
Wave…, 2003; Stopa et al., 2016; Liu Q. et al., 2016), a significant part of the year stormy weather prevails over the
Barents Sea. The duration of periods in which the wind speed does not exceed 15 m/s in the winter months averages
only 3–6 days. The mean wave height (probability of exceedance 50%) with a frequency of occurrence of 1 time per
year is 6.1 m, and the maximum wave height (probability of exceedance 0.1%) is more than 19 m (Wind and Wave…,
2003). Such values indicate the high frequency of occurrence of extreme waves. The average significant wave heights
of in the Barents Sea is 1.8–2.2 m for the central part of the Barents Sea (Myslenkov et al., 2019). The maximum of
significant wave heights reaches 12–14 m in the central part of the Barents Sea. The storms with significant wave
heights of more than 4 m are observed on average 70–80 times a year, with significant wave heights more than 5 m -
40–60 times a year. The interannual variability of the recurrence of storm waves is very large (for different years the
number of cases can vary by a factor of 2–3) (Myslenkov et al., 2018, 2019).
Moreover, the wave climate of the Barents Sea is characterized by a significant influence of swell coming
from the North Atlantic. Based on numerical experiments (Myslenkov et al., 2015), it was shown that the height of
swell can reach 5 m with a period of 15-18 sec. The effect of swell is not taken into account in the Charnock
relationship explicitly, which can cause errors in the calculated values of the roughness length and turbulent fluxes.
In addition to wind speed, the difference of temperature and specific humidity between the sea surface and air
also affects the magnitude of turbulent heat fluxes over the sea. These differences reach particularly large values
during the so-called cold-air outbreaks (CAOs). CAOs represent the advection of a dry and cold air mass onto the
open sea originating from the Central Arctic or from the cold continents (Pithan et al., 2018). The temperature
difference between water and air during CAOs can exceed 30 °C near the marginal sea ice zone, and the maximum
values of the total turbulent heat flux can exceed 600 W/m$^2$ (Brümmer, 1996). As the air mass warms and moistens
with increasing distance from the ice edge, the total heat flux decreases. The horizontal scale of the air mass
transformation is about 500-1000 km for typical CAOs (Chechin and Lüpkes, 2017). Thus, large areas of the non-
freezing seas, such as the Barents Sea, are subject to intense heat loss. The heat loss due to CAOs can reach up to 60%
over the Greenland and Iceland Seas (Papritz and Spengler, 2017), although the specific value depends on the criteria
used for the identification of CAOs. To our knowledge, no systematic study of the CAOs role in the air-sea heat



exchange exists for the Barents Sea, although the importance of CAOs has been stressed earlier (Smedsrud et al.,
2013).

Furthermore, CAOs create favorable conditions for enhancing wind speed over water, which leads to further
intensification of the energy exchange. The wind speed increase is primarily associated with the formation of large
horizontal temperature gradients and strong baroclinicity. This can lead to the intensification of cyclones and
mesocyclones (Kolstad, 2015), formation of jets and wind shear along the lower tropospheric fronts (Grønas and
Skeie, 1999), convergence lines (Savijärvi, 2012), and low-level jets (Brümmer 1996; Chechin et al., 2013; Chechin
and Lüpkes, 2019). Although the highest wind speeds over the Barents Sea have the orographic origin (e.g., the
Novaya Zemlya Bora (Moore, 2013)), it was shown (Kolstad, 2015) that in cyclones, the wind speed reaches its
maximum value when intense cold advection takes place in their rear part. In addition, intense turbulent exchange in
the convective boundary layer effectively transports momentum down to the lower atmospheric layer increasing the
near-surface wind speed (Chechin et al., 2015).

In this paper, we consider the influence of sea waves on turbulent heat fluxes in the Barents Sea. Heat fluxes
were calculated using the COARE 3.0 algorithm and NCEP/CFSR reanalysis data with the Charnock roughness
length parameterization and parameterizations explicitly taking into account the parameters of sea waves - Taylor and
Yelland (2001), Oost et al. (2002) and Drennan et al. (2003). The results was verified by the ship measurements of
turbulent heat fluxes obtained during the NABOS (Nansen and Amundsen Basins Observational System) campaigns
in different years. The wind wave parameters were obtained from the WaveWatchIII (WWIII) wave model. Special
attention is paid to the cases of intense storms and cold-air outbreaks events, when the expected difference between
calculations with different roughness parameterizations is the largest.

**2. Data and Methods**

**2.1 Wave modeling**

The wave characteristics in the Barents Sea were computed using the spectral wave model WaveWatchIII
(WWIII) version 4.18. The WW3 model is an development of the WAM model with regard to the functions of the
source and the nonlinear interaction (Tolman, 2014). This model is based on a numerical solution of the equation of
the spectral wave energy balance
$$\frac{\partial E(\omega, \theta, \vec{x}, t)}{\partial t} + \vec{V}(\omega, \theta) \nabla E = S(\omega, \theta, \vec{x}, t), \tag{1}$$

where $\omega$ and $\theta$ are the frequency and the propagation direction of the spectral component of the wave energy;
$E(\omega, \theta, \vec{x}, t)$ is the two-dimensional spectrum of the wave energy at a point with vector coordinate $\vec{x}$ at time
point $t$; $\vec{V}(\omega, \theta)$ is the group velocity of the spectral components; $S(\omega, \theta, \vec{x}, t)$ is a function that describes
the wave energy sources and sinks, i.e., the transfer of the energy from the wind to the waves, nonlinear wave
interactions, dissipation of the energy through collapse of the crests at a great depth and in the coastal zone, friction
against the bottom and ice, wave scattering by ground relief forms, and reflection from the coastline and floating
objects. The energy balance equation is integrated using finite-difference schemes by the geographic grid and the
spectrum of wave parameters.
In this work, the computations were made using the ST1 scheme (Tolman, 2014). To account for the
nonlinear interactions of the waves, the Discrete Interaction Approximation (DIA) model was used, which is a
standard approximation for calculation of nonlinear interactions in all modern wave models.





To take into account ice effects on the wave development, the IC0 scheme was used, where the grid point is
considered as ice-covered if the ice concentration was larger than 0.25. Thus, the exponential attenuation of wave
energy adjusted for the sea ice concentration at a given point was added.
In the shallow water, the increase in wave height as waves approach the shore and the related wave breaking
after waves reach the critical value of steepness were taken into consideration. The whitecapping effect taken into
account in the ST1 scheme. The standard JONSWAP scheme was used to take the bottom friction into account. The
spectral resolution of the model is 36 directions (Dq = 10°), the frequency range consists of 36 intervals (from 0.03 to
0.843 Hz).
The calculations were performed using the original unstructured grid, which is based on the bottom
topography data from ETOPO1 database and detailed nautical charts (Figure 1). This unstructured grid consists of
16792 nodes; the spatial resolution varies from 15 km for the open part of the Barents Sea to 500 m for the coastal
regions. The computational domain of the model covers the Barents and the Kara Seas and the entire northern part of
the Atlantic Ocean (Figure 1). Previously, this grid was successfully used for wave modeling (Myslenkov et al., 2018;
Myslenkov et al., 2019). The need to take into account the swell propagating from Atlantic ocean when calculating the
height of significant waves in the Barents Sea was clearly shown in the previous work of the authors (Myslenkov et
al., 2015).

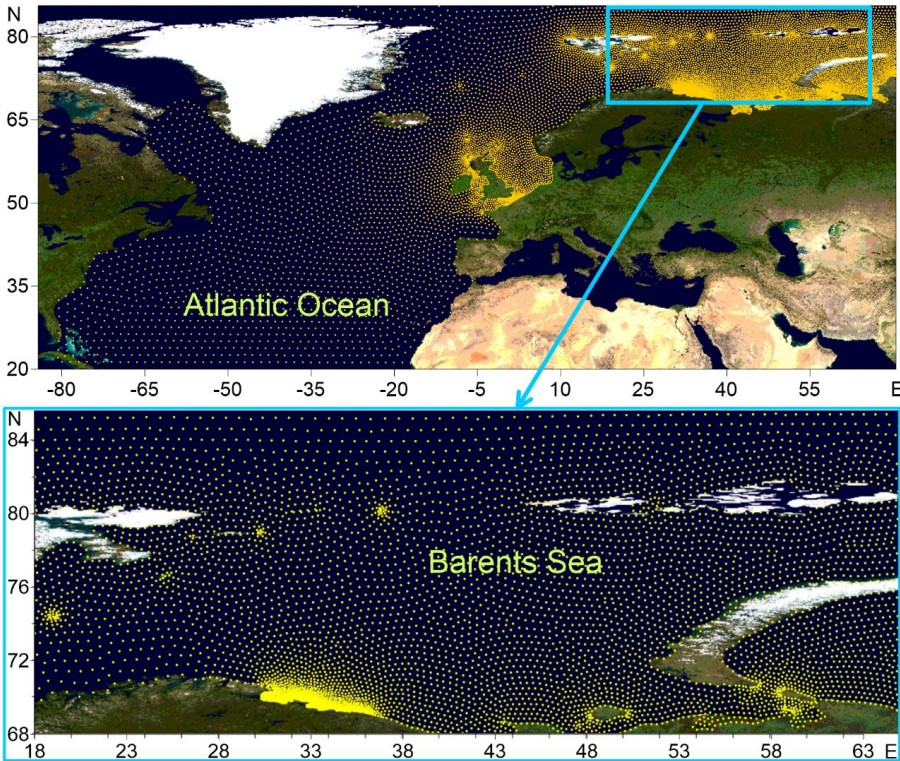


Figure 1. The computational unstructured grid for the Atlantic Ocean and the Barents Sea. The base map is
the Blue Marble which obtained by connecting to the WMS demo server in the Surfer Golden Software program.

The general time step for the integration of the full wave equation was 15 minutes, the time step for the
integration of functions of sources and sinks of wave energy was 60 s, the time step for the spectral energy transfer


and for satisfying the Courant–Friedrichs–Lewy condition was 450 s. This choice is dictated by the configuration of
the computational grid: the maximum and minimum distances between the nodes and a large latitudinal extent.
The 10-m wind from the NCEP/CFSR reanalysis (Saha et al., 2010) for the period of 1979 to 2010 with the
spatial resolution of ~0.3° was used as the forcing. Data of NCEP/CFSv2 reanalysis (Saha et al., 2014) with the
resolution of ~0.2° and with the time step of 1 hour were used for the period of 2011 to 2017.
In this paper, we used the output results of the wave model with time step 3 hours from 1979 to 2017 for each
node of the unstructured grid.
Based on the wave model results, a study of storm activity was carried out according to the POT (Peak Over
Threshold) method which used successfully earlier in (Myslenkov et al., 2019). For each year in the Barents Sea, the
number of storm surges with different significant wave heights from 5 to 8 m was calculated. The event is counted as
the storm with wave height > 5 m if at least in one node in the study area the wave height exceeds the threshold of 5 m
This event continues until the wave height at all nodes becomes less than the threshold. To eliminate possible errors,
at least 9 hours should pass between two storm events. Using the described procedure, a catalog of storm days was
compiled when the significant wave heights of more than 5 m were observed. A total of 1964 days were identified for
the period 1979-2017.

**2.2 COARE algorithm and parameterizing the roughness parameter**
Turbulent heat fluxes were calculated using the COARE algorithm (Fairall et al., 1996), based on the LKB
model (Liu et al., 1979). Bulk formulae for the momentum and scalar fluxes have the general form:

$$w'x' = c_x^{1/2} c_d^{1/2} S\Delta X = C_x S\Delta X, \tag{2}$$

where $x$ is the horizontal wind components $u$, $v$, temperature or specific humidity, $c_x$ – transfer coefficients
for $x$, $c_d$ – transfer coefficient for momentum, $C_x$ – total transfer coefficient, $\Delta X$ – the difference the mean $x$ at a height
equal to the roughness length and at a certain height (10 m) in the atmospheric surface layer (Fairall et al., 2003). $S$ –
mean wind speed with gusts $U_g$:

$$S = \sqrt{U^2 + V^2 + U_g^2}$$

The default value of $U_g$ is 0.5 m/s in the COARE algorithm. Transfer coefficients depend on the roughness
length and dimensionless universal functions. The form of universal functions in the COARE algorithm is set in
accordance with (Beljaars and Holtslag, 1991) for stable stratification; the so-called Kansas functions (Kaimal et al.,
1972) are used for unstable stratification; functions from Fairall et al. (1996) and Grachev et al. (2000) are used for
very unstable stratification. For the roughness length, several parameterizations are available in the COARE
algorithm. The parameterization of Charnock (Charnock, 1955) implies dependence of roughness on the friction
velocity $u_*$:

$$z_0 = \frac{\alpha u_*^2}{g} + \frac{0.11a}{u_*} \tag{3}$$

where $\alpha$ – Charnock parameter, $g$ – gravity acceleration, $a$ – kinematic viscosity coefficient (Andreas, 1989).
Equation (3) is the modified Charnock formula (Smith, 1988), in which the second term on the right side describes the
roughness over an aerodynamically smooth surface (i.e., in weak winds). The Charnock coefficient is set piecewise
constant in strong and weak winds and linearly dependent on 10-m wind speed in moderate winds:

$$
\quad
\begin{cases}
0.011, & S < 10 \; m/s \\
0.011 + \dfrac{0.007(S - 10)}{8}, & 10 \; m/s < S < 18 \; m/s \\
0.018, & S > 18 \; m/s
\end{cases}
$$






In the parameterization of Taylor and Yelland (2001) (hereafter - T1), the roughness length is related to the
wave steepness ($H_s/L_p$):
$$z_0 = H_s a_1 \left(\frac{H_s}{L_p}\right)^{b_1} + \frac{0.11a}{u_*}, \quad a_1 = 1200, \quad b_1 = 4.5 \tag{4}$$
where $H_s$ – significant wave height, $L_p$ – spectral peak wavelength.
The parameterization of Oost et al. (2002) (hereafter - O2) implies the dependence of the roughness length on
the spectral peak wavelength $L_p$ and inverse wave age ($U_*/c_p$):
$$z_0 = L_p a_2 \left(\frac{u_*}{c_p}\right)^{b_2} + \frac{0.11a}{u_*}, \quad a_2 = 50/2\pi, \quad b_2 = 4.5 \tag{5}$$
Here $c_p$ –phase wave speed associated with spectral peak, which is expressed through the wave length as
$c_p = \sqrt{L_p g/2\pi}$ .
Finally, we included the parametrization of Drennan et al. (2003) (hereafter - D3) in the COARE algorithm.
D3 parameterization consists in the dependence of the roughness length on the wave height and inverse wave age:
$$z_0 = H_s a_3 \left(\frac{u_*}{c_p}\right)^{b_3} + \frac{0.11a}{u_*}, \quad a_3 = 3.35, \quad b_3 = 3.4 \tag{6}$$
Thus, the main components of the algorithm are the equation (2), formulae for calculating transfer
coefficients based on the Monin-Obukhov similarity theory, and formulae (3-6) for the roughness length. Thus, in
general, the COARE algorithm is similar to corresponding algorithms in most atmospheric models.
Using the COARE algorithm, we calculated turbulent sensible and latent heat fluxes in the Barents Sea from
1979 to 2017. Mean fluxes were calculated for long-term period and for periods of cold-air outbreaks and storm wave
events.

**2.3 Input data for the COARE algorithm**
Input data for the COARE algorithm are: wind vector, air temperature, sea surface temperature (SST), air
humidity, incoming short-wave and long-wave radiation, precipitation intensity, sea wave height and period.
NCEP/CFSR and CFSv2 (Saha et al., 2010, 2014) reanalysis with temporal resolution of 6 hours and total period
1979-2017 were used as atmospheric data input for the COARE algorithm . CFSv2 reanalysis data for the period
2011-2017 (with a slightly better spatial resolution than CFSR, were interpolated from the ~0.2˚ grid to ~0.3˚ grid to
match the CFSR resolution. The wind speed was used at 10 m height, air temperature and humidity were used at 2 m
height. Reanalysis data are also available at isobaric levels, the lower of which is 1000 hPa. However, we preferred to
take diagnostic variables at heights of 2 and 10 m for several reasons. Firstly, the height of the isobaric levels varies
greatly and the lower available level may be at a high height (above the boundary layer). Secondly, data at vertical
levels are available on a much coarser grid (0.5 °). For instance, Arthun and Schrum (2010) also used diagnostic
variables at standard levels from the NCEP-NCAR reanalysis to calculate turbulent fluxes in the ocean model. The
surface pressure and the inversion height (boundary layer height), which are usually set constant in the COARE
algorithm, were set from the CFSR reanalysis (at each moment of time and at each grid point).

**2.4 Ship observations**



We used ship observations in the Barents Sea from the NABOS expeditions in 2005, 2007, 2013, and 2015 to

verify turbulent heat fluxes calculated using the COARE algorithm. All expeditions took place in a period from
August to October. Ship-borne fluxes were calculated using the eddy-covariance method (the left side of equation (2))
based on high-frequency measurements of temperature and the three wind components using Gill and Metek sonic
anemometers (Ivanov et al., 2019; Varentsov et al., 2016). The averaging period for the covariance calculations was
10 min. For all wind measurements, a correction was made for the movement of the ship. A detailed description of the
location of the instruments and methods of filtering data and calculating fluxes is available at https://uaf-
iarc.org/nabos-cruises/. For verification, the calculated values of heat fluxes were bilinearly interpolated from the
CFSR reanalysis grid to the observation points.

**2.5. Identification of CAOs**

The so-called «CAO index» is frequently used for CAO identification. It was first defined (Kolstad and

Bracegirdle, 2008; Kolstad et al., 2009) as the potential temperature difference between the ocean surface and the 700
hPa height normalized by the pressure difference at the same heights. The authors used the value of the 90th
percentile of the CAO index to estimate the strength and frequency of occurrence of CAOs. Other investigators (e.g.,
Fletcher et al., 2016) used the non-normalized potential temperature difference between the surface and the 800hPa
height. As metrics to study the frequency and strength of CAOs they evaluated the frequency of occurrence of the
positive values of the CAO index, as well as the value of the 95th percentile of the CAO index during the winter
months.

Here, we define the CAO index $I_{cao}$ as the daily potential temperature difference between the ocean surface

and the 700 hPa height. For each day, $I_{cao}$ was averaged over the ice-free part of the Barents sea. Figure 2 shows the
obtained $I_{cao}$ values for the period 1979-2018. Solid curve on Figure 2 consists of the multiyear-averaged values
$\overline{I_{CAO}}$ obtained by 1) averaging $I_{cao}$ over a 30-day period centered on the given day and 2) averaging the obtained
values over the years. Similarly, the standard deviation $\sigma_I$ of $I_{cao}$ was obtained.



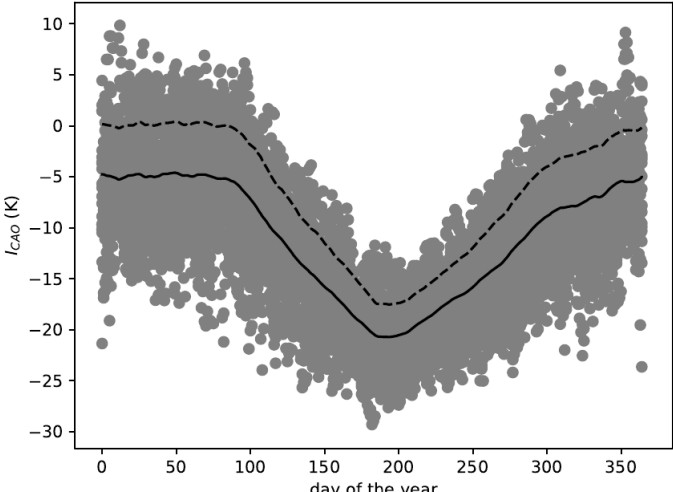

Figure 2. Cold-air outbreak index $I_{cao}$ for the period 1997-2017. Solid curve represents the 30-day running
multiyear mean values $\overline{I_{CAO}}$. Extreme CAOs correspond to points above the dashed curve which is the sum $\overline{I_{CAO}}$
$+ \sigma_I$ where the latter is the 30-day running multiyear standard deviation of $I_{cao}$.

The dashed curve in Figure 2 represents the threshold value $\overline{I_{CAO}} + \sigma_I$ which we use as a criteria for CAO
identification, namely
$$I_{CAO} > \overline{I_{CAO}} + \sigma_I \qquad (7)$$
According to the criteria (7), we identify CAOs as those cases when $I_{cao}$ values are above the dashed curve in
Figure 2. A similar procedure was used in other studies (e.g., Wheeler et al., 2011) to identify continental CAOs
where authors used simply the air temperature at 2 m height instead of $I_{cao}$.
Figure 2 shows that the largest values of $I_{cao}$ are observed in a period from the second half of December until
the end of March when the coldest air advection occurs over the Barents Sea. It is interesting to note that in winter the
criteria (7) is almost identical to simply $I_{cao} > 0$. The latter serves as a measure of the dry hydrostatic stability of the
layer between the ocean surface and the 700 hPa surface. Thus, positive values of $I_{cao}$ indicate conditions favorable for
the mixed-layer development to the heights over 700 hPa. During strong background advection mixed-layer can reach
such heights only at a significant distance from the ice edge (Chechin and Lüpkes, 2017).

**3. Results**
**3.1 Wave climate and storm activity**
First, we consider the main features of wave conditions and wave climate in the Barents Sea, which directly
affect the processes of heat exchange in the ocean-atmosphere system. In Figure 3 the average significant wave
heights for the entire simulation period from 1979 to 2017 is shown. The highest average wave heights are found in
the western part of the sea. Here we can expect the greatest influence of sea waves on heat fluxes. In the north, due to
the presence of ice, the average wave heights do not exceed 1 m.

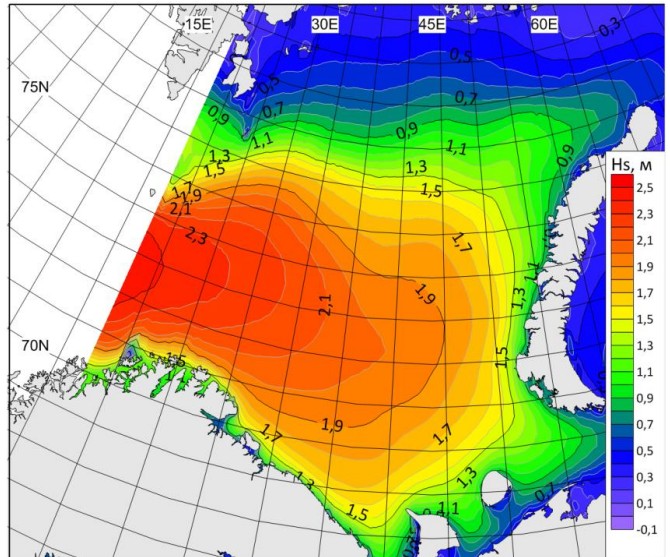


Figure 3. Long-term average significant wave height in the Barents Sea based on the WWIII simulation
results for the period 1979-2017.

Also, an equally important parameter is the wavelength, which is used in the parametrizations O2 and D3. In
Figure 4 the mean long-term spectral peak wavelength is shown. The wavelengths 80-100 m are observed in the
central and western parts of the Barents Sea. The results on the average wave height and wavelength in general are
consistent with similar works by other authors (Semedo et al., 2011; Stopa et al., 2016). Estimates of storm activity
based on such long-term analysis are relatively rare and their detailed analysis is necessary an additional research.

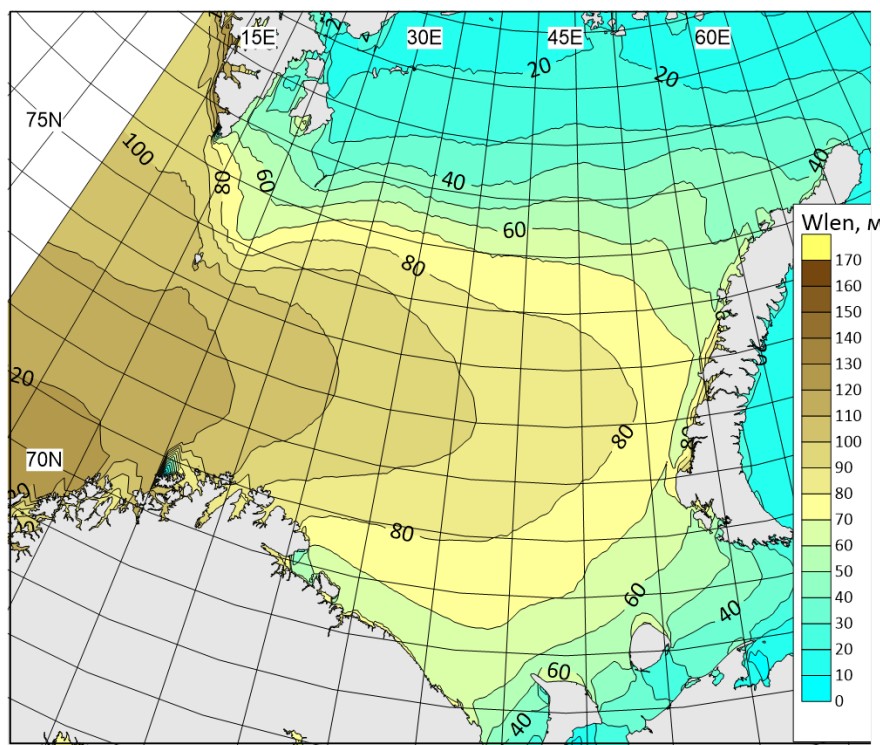

Figure 4. Long-term average long-term spectral peak wavelength in the Barents Sea based on the WWIII
simulation results for the period 1979-2017.

The Barents Sea is characterized by a high frequency of storm wave events, which provide a long swell in the
extinction stage (i.e., "old seas") and limit the applicability of the Charnock formula. As shown in (Myslenkov et al.,
2018), the number of storms per year in the Barents Sea can differ significantly. Figure 5 shows the number of storms
calculated according to the wave model results with wave heights of more than 5 m and more than 7 m (identified as
described in the Section 2.1). During the period from 1979 to 2017, several maxima of storm activity were observed,
for example, in 1989-1991 and in 2011. Especially for these periods, the calculated heat fluxes are expected to be
sensitive to the used of parameterizations of the roughness length (see Section 3.5).

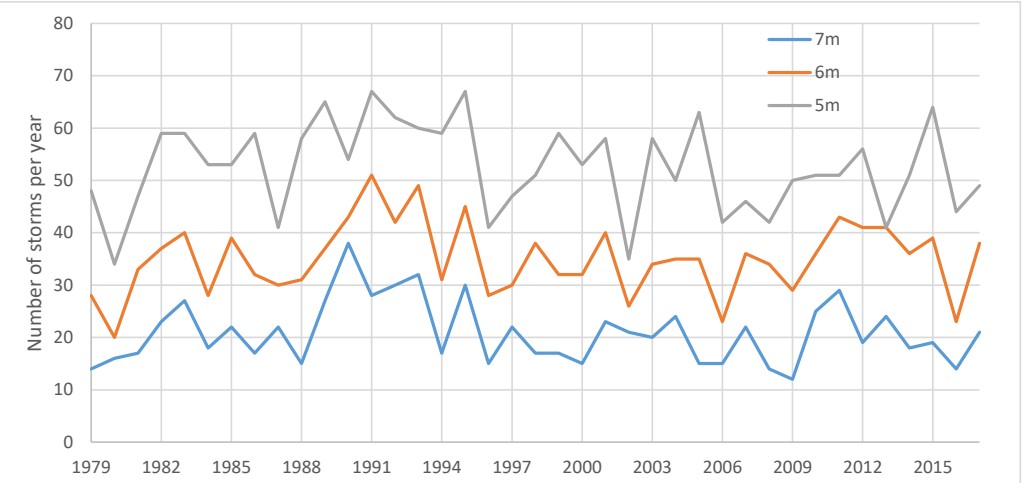

Figure 5. The number of storms with a significant wave height of more than 5, 6 and 7 m according to the WWIII simulation results for the period 1979-2017.

### 3.2 CAOs frequency of occurrence

Figure 6 shows the timeseries of the number of days with extreme CAOs selected using criteria (7) for each cold period (November-April) of 1979-2018. On average, CAOs are observed in 16.4 % days. However, the interannual variability of the frequency of occurrence of CAOs is large. Namely, the interannual standard deviation of the number of CAO days amounts to 12 days. Thereby, the number of CAO days per cold season varies from 6 in 2011-2012 to 56 in 1980-1981.

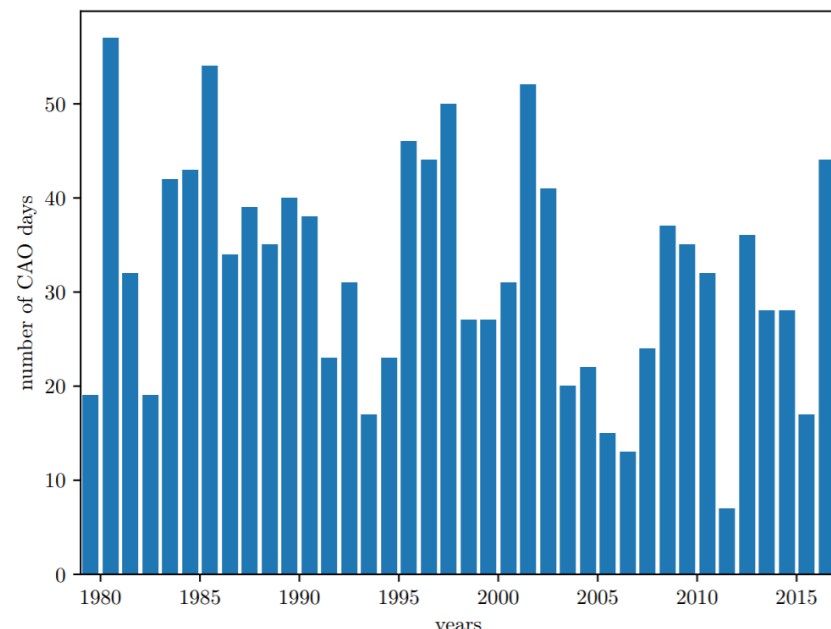

Figure 6. The number of days with CAOs over the Barents Sea selected using criteria (7) for each cold season in 1979-2018.





The frequency of occurrence of CAOs over the Barents Sea is governed by the variability of the largescale
patterns of atmospheric circulation. To the largest extent, the frequency of CAOs is correlated with the so-called
«Barents Oscillation» (Skeie et al., 2000; Wu et al. 2006; Kolstad et al., 2009). The latter is the mode of variability of
the sea-level pressure field represented by a dipole with high pressure over Greenland and Iceland and low pressure
over the northern part of the European part of Russia. Such pressure field promotes intense cold-air advection over the
Barents Sea from the north. Moreover, there is a negative correlation between the North Atlantic Oscillation index and
CAOs frequency of occurrence (Kolstad et al., 2009). Such a correlation is particularly strong for easterly CAOs,
which is obviously associated with the reduced strength of the westerlies.

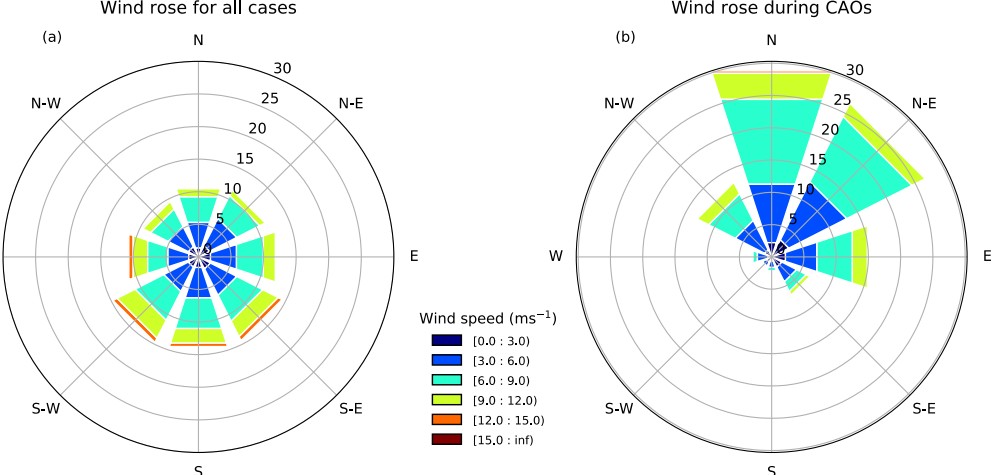

Figure 7. Frequency of occurrence of daily 10 m wind speed and direction, averaged over the ice-free part of
the Barents Sea for the period November-April 1979-2018 for all cases (a) and cold-air outbreaks (b).

The frequency of CAOs with easterly wind over the Barents Sea is significant and amounts to 16% of all
CAOs (Figure 7b). During CAOs, the highest frequency of occurrence have northerly (30%) and north-easterly (27%)
winds. The wind rose in CAOs differs from the wind rose in all cases during the cold season (Figure 7a). In particular,
the prevailing wind direction over the Barents sea in winter is from the south. Moreover, the winds having southerly
and westerly components are the strongest.
The CAOs role in the heat exchange between the Barents Sea and the atmosphere is demonstrated by Figure
8. The latter shows the turbulent fluxes of sensible and latent heat, $H$ and $LE$, respectively, the net longwave radiative
flux $LW_{net}$, and the total heat flux $F_{total} = H + LE + LW_{net}$ averaged over the November-April period over the ice-
free part of the Barents Sea as functions of the number of CAO days during the same period. Clearly, there is a strong
dependency of the Barents Sea on the number of CAO days. The highest correlation coefficients are obtained for
$LW_{net}$, $F_{total}$ and $H$ amount to 0.86, 0.85 and 0.84, respectively. A smaller correlation coefficient of 0.78 is obtained
for $LE$. Also, the coefficients of linear regression shown in Figure 8 demonstrate that $F_{total}$ has the strongest
dependency on the number of CAO days. From all terms of the surface heat balance, the sensible heat flux $H$ is most
sensitive to the number of CAO days. All the three considered components of the surface heat balance ($H$, $LE$ and
$LW_{net}$) manifest heat loss from the sea surface to the atmosphere and are of comparable magnitude of about 70 Wm$^{-2}$
on average.


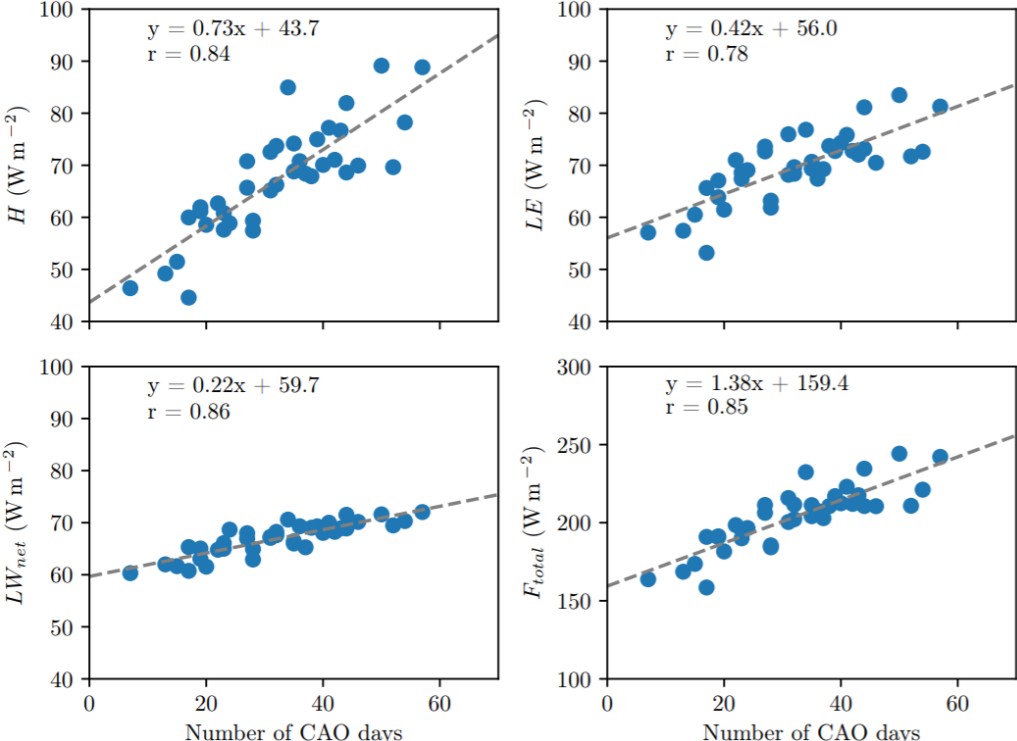


Figure 8. Turbulent fluxes of sensible and latent heat, $H$ и $LE$ respectively, net longwave radiative flux $LW_{net}$ and the total heat flux $F_{total} = H + LE + LW_{net}$ averaged over the cold season (November-April) and over the ice-free part of the Barents Sea as function of number of CAO days during the same period for 1979-2018. Dashed line shows the linear regression line, whose equation is given at each plot, as well as the correlation coefficient $r$.

376   We stress that the values of fluxes shown in Figure 8 are averaged over the ice-free part of the Barents Sea. It
377 is important to keep in mind that there is a large interannual variability of the area of sea ice cover in the Barents Sea.
378 This is another important factor, along with the number of CAO days, influencing the heat loss.

380   **3.3 Verification of the COARE algorithm by the ship observations**
381  Figure 9 shows the comparison of sensible and latent heat fluxes from shipborne observations and calculated
382 using different roughness parameterizations, namely Charnock, 1955 (C55), Taylor and Yelland, 2001 (T1), Oost et
383 al., 2002 (O2) and Drennan et al., 2003 (D3).



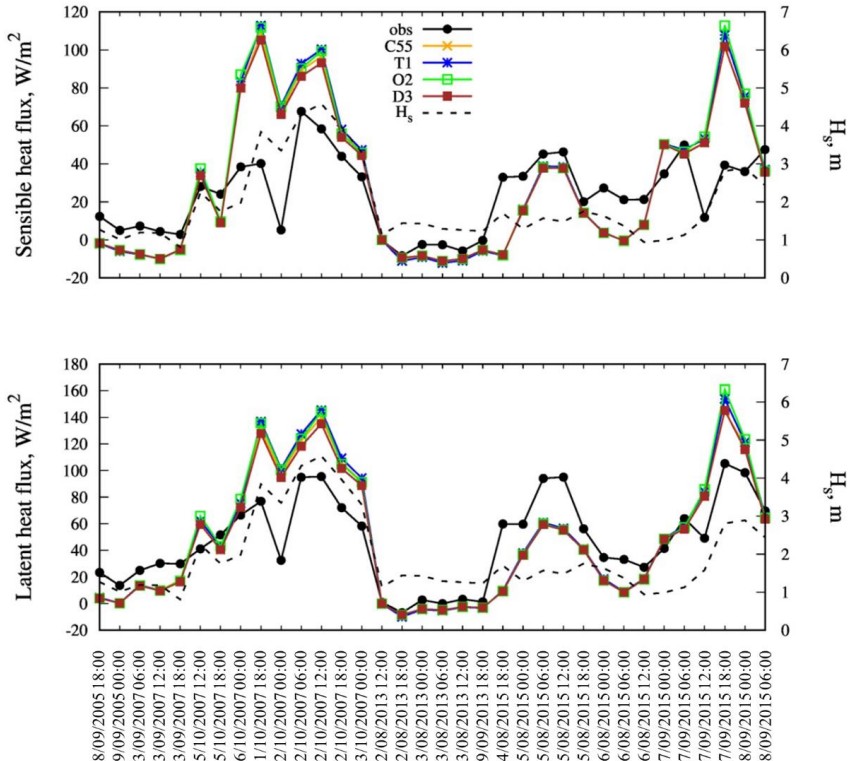


Figure 9. Sensible (top) and latent (bottom) heat fluxes according to NABOS observations and calculated
using various roughness parameterizations. The dashed line shows the significant wave height $H_s$ according to WWIII
simulations.

Heat fluxes calculated with different roughness parameterizations are almost identical (Figure 9); an average
difference between them is 1 W m$^{-2}$. The correlation coefficient between the observed and the calculated fluxes is 0.7
for the sensible heat flux and 0.8 for the latent heat flux. However, the mean absolute error (MAE) is rather large -
about 20 W m$^{-2}$. The error magnitude increases with the increase of the heat flux magnitude. The error may be
connected both with the COARE algorithm itself and with the input data (i.e., related to the quality of meteorological
parameters in the reanalysis). For example, a strong overestimation of heat fluxes on October 11–12, 2007 is
associated with the overestimation of wind speed (by 6–8 m s$^{-1}$) compared to observations.
In order to estimate the accuracy of the COARE algorithm itself, we excluded from the analysis those
samples for which the reanalysis errors were large, namely, when the wind speed error exceeded 4 m s$^{-1}$ and/or the
SST and air temperature error exceeded 1.5 °C and/or the error in specific humidity exceeded $0.7 \cdot 10^{-3}$ kg kg$^{-1}$. For
such a sample, MAE decreased by half, to 10 W m$^{-2}$. For those periods when the error of wind speed, temperature, and
specific humidity was the smallest, an error in the resulting heat fluxes also becomes small and amounts to about 5 W
m$^{-2}$, which is most likely associated with the inaccuracy of the COARE algorithm. However, this error is within the
accuracy of the eddy-covariance method. The accuracy of this method in the case of ship measurements can be
significantly reduced due to the influence of air flow distortion by the ship. Therefore, we can conclude that the





calculated fluxes are in good agreement with the observations. It should be noted that the error between the observed
and calculated fluxes for all parameterizations exceeds the difference between calculated fluxes using different
parameterizations.

**3.4 Long-term mean turbulent heat fluxes**

Here we consider the mean long-term values of heat fluxes calculated from the CFSR reanalysis data using

COARE algorithm and various roughness parameterizations. The mean (for the period 1979-2017) sensible and latent
heat flux in the experiment C55 and difference in the between different experiments shown on Figure 10, 11. The
main conclusion of these results is the presence of positive difference for T1 and O2 experiments and negative for D3.
The long-term values of difference is small: 1-2 W m$^{-2}$ for T1 and 0.5-1 W m$^{-2}$ for O2.

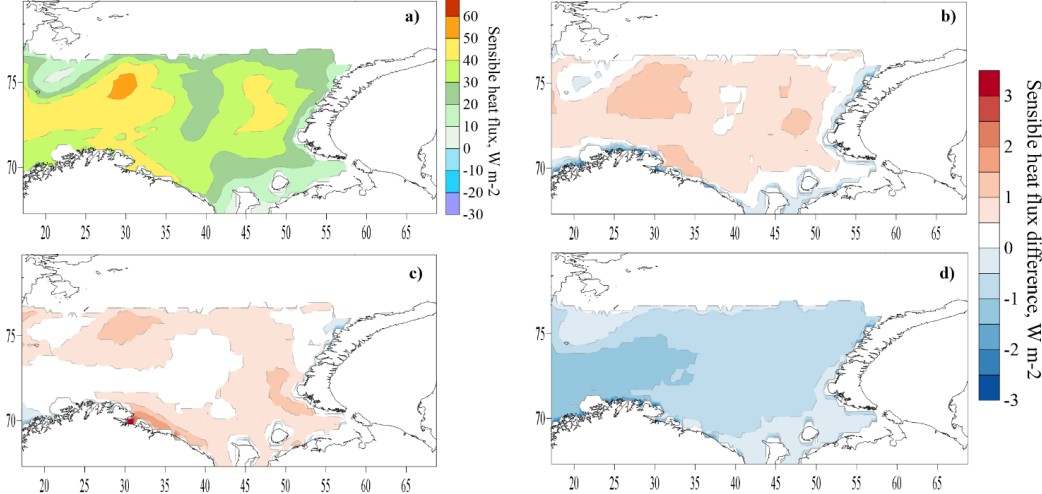


Figure 10. Mean sensible heat flux in the experiment C55 (a,) and the difference in the sensible heat fluxes

between experiments T1 - C55 (b), O2 - C55 (c) and D3 - C55 (d). All grid nodes where sea ice was in more than half
of the cases are filtered.

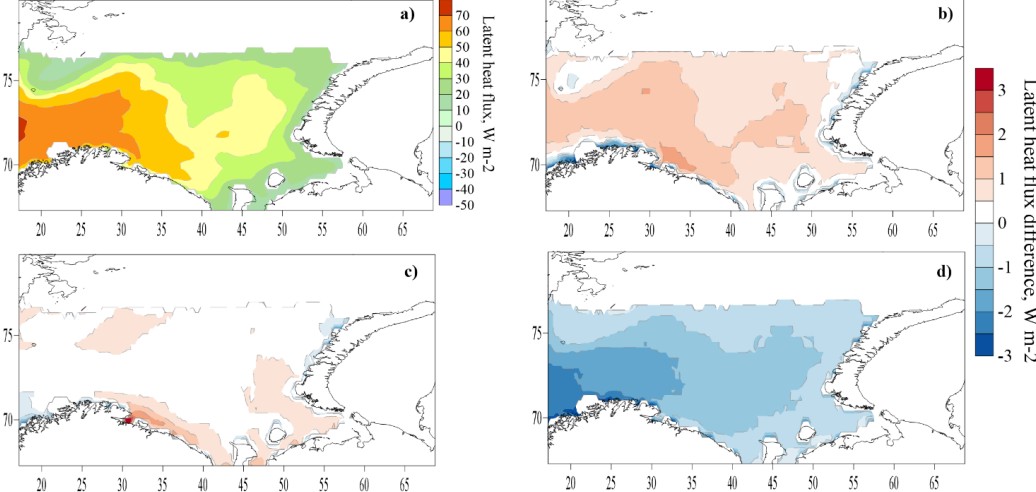






419   Figure 11. Mean latent heat flux in the experiment C55 (a,) and the difference in the latent heat fluxes

420 between experiments T1 - C55 (b), O2 - C55 (c) and D3 - C55 (d). All grid nodes where sea ice was in more than half

421 of the cases are filtered.


423   Tables 1, 2 show the average statistics: the difference in heat fluxes with and without explicit accounting for

424 sea waves parameters. Over the entire Barents Sea, the full range of differences in the fluxes are small, within $-3 \div 2$

425 W m$^{-2}$, which is only 1-3% of the mean absolute value. The greatest mean difference for sensible heat flux observed

426 for T1 and for latent heat flux for O2 parametrization.

427   The flux difference can exceed 30-50 W m$^{-2}$ (in 0.1% of cases or 99.9 percentile) and in some extreme cases

428 reach 100-250 W m$^{-2}$. The highest maxima of the flux difference are obtained for the experiment O2.

429

430 **Table 1**

431   Statistical characteristics of the difference in the sensible heat flux calculated with and without explicit

432 accounting for sea waves parameters: mean difference, relative mean (ratio of the mean difference to the mean value

433 of the flux), mean absolute difference, 95 and 99.9 percentile and the maximum difference for the Barents Sea

|  | Mean difference (W m$^{-2}$) | Relative mean difference (%) | Mean absolute difference (W m$^{-2}$) | 95 percentile (W m$^{-2}$) | 99.9 percentile (W m$^{-2}$) |
|---|---|---|---|---|---|
| T1 - C55 | 0.5 | 1.4 | 1.7 | 7.3 | 40 |
| O2 - C55 | 0.6 | 2.1 | 1.6 | 6.7 | 56 |
| D3 - C55 | -0.7 | -2.3 | 1.1 | 3.7 | 35 |

434

435 **Table 2**

436   Statistical characteristics of the difference in the latent heat flux calculated with and without explicit

437 accounting for sea waves: mean, relative mean (ratio of the mean difference to the mean value of the flux), mean

438 absolute difference, 95 and 99.9 percentile and the maximum difference for the Barents Sea

|  | Mean difference (W m$^{-2}$) | Relative mean difference (%) | Mean absolute difference (W m$^{-2}$) | 95 percentile (W m$^{-2}$) | 99.9 percentile (W m$^{-2}$) |
|---|---|---|---|---|---|
| T1 - C55 | 0.7 | 1.6 | 1.8 | 6.7 | 41 |
| O2 - C55 | 0.6 | 1 | 1.7 | 6.4 | 50 |





| | | | | | |
|---|---|---|---|---|---|
| D3 - C55 | -1.1 | -2.8 | 1.3 | 3.7 | 38 |


The greatest differences between the experiments are found in those areas where the highest values of the
heat fluxes are observed. This can be explained by the power-law dependence of the roughness length on the friction
velocity / wave height. Moreover, in the O2 parameterization, the proportionality coefficient is larger (a2 = 4.5) than
in the D3 parameterization (a3 = 3.4), which is reflected in the flux differences.
A more detailed spatial analysis of 99.9 percentile of sensible heat flux difference shown on Figure 12. The
extreme values of the flux difference taking O2-C55 difference as an example showed that some of the extrema are
associated with coastal areas, mainly off the western coast of Novaya Zemlya during bora. Other extremes were
associated with deep cyclones in different parts of the sea, with different distances from the coast. Some extremes are
associated with storm waves or are observed immediately after storms, during cold-air outbreaks in the rear of
cyclones. Therefore, the characteristics of heat fluxes during storm waves and cold-air outbreaks will be considered
separately in the following sections.

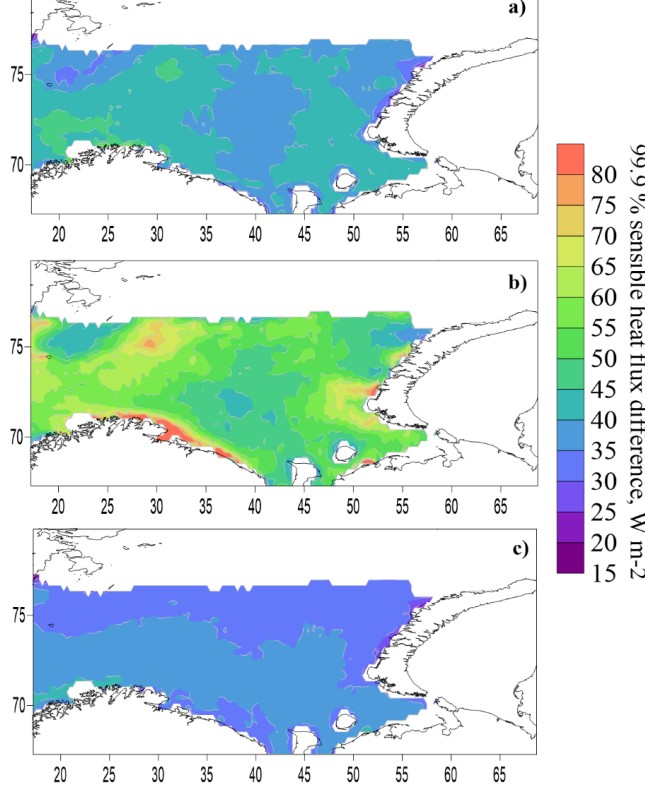


Figure 12. 99.9 percentile of sensible heat flux difference between experiments T1 - C55 (a), O2 - C55 (b)
and D3 - C55 (c)

**3.5 Turbulent heat fluxes during storm wave events**
Here we consider turbulent heat fluxes during the storms identified in Section 3.1 (a total of 1964 days with
storms for the period 1979-2017). The spatial distribution of heat fluxes during storms (Figure 13, 14) resembles the





average distribution (Figure 10, 11), but the absolute values increase by almost a factor of 2. The average sensible
heat flux has several maxima - in the northwest of the sea, near the coast of the Kola Peninsula and a less pronounced
local maximum off the southern island of Novaya Zemlya. The flux difference between the experiments is also
distributed the same as on average and increases in absolute value (except for experiment D3). The average flux
difference between experiments reaches 4-5 W m$^{-2}$ for T1-C55, 8 W m$^{-2}$ for O2-C55 and 3-4 W m$^{-2}$ for D3-C55. On
average, the relative difference in heat fluxes is 3% for T1-C55 and 3-5% for O2-C55. The correlation coefficient
between the magnitude of the flux and the magnitude of the flux difference is 0.9. For the D3 experiment, the flux
difference gradually increases from east to west, and some special structure associated precisely with storms does not
appear. The detected maxima of flux difference in the western part of Sea generally correspond to the maxima of the
average wave height (Figure 3).
It can be concluded that the mean pattern of heat fluxes in the Barents Sea is largely contributed by storms.

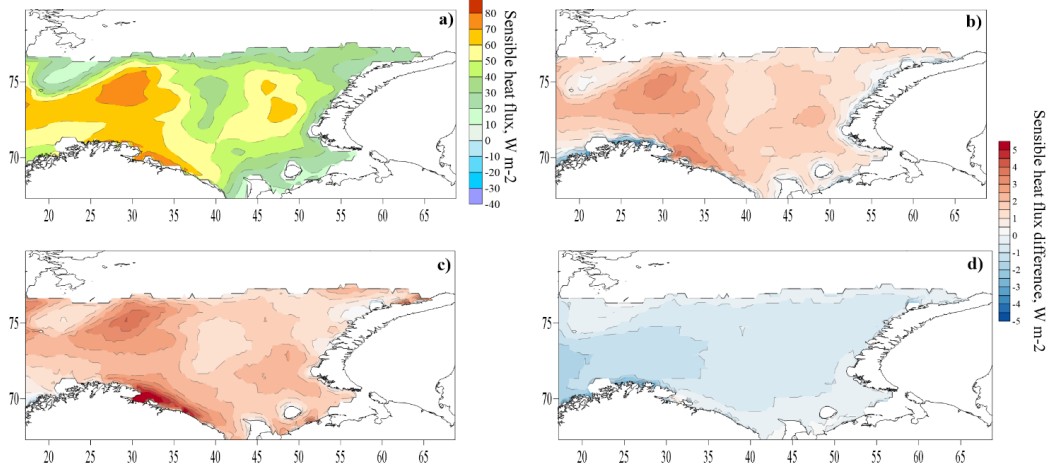


Figure 13. Mean sensible heat flux in experiment C55 (a) and the flux difference in experiments T1 - C55
(b), O2 - C55 (c) and D3 - C55 (d) during storms.

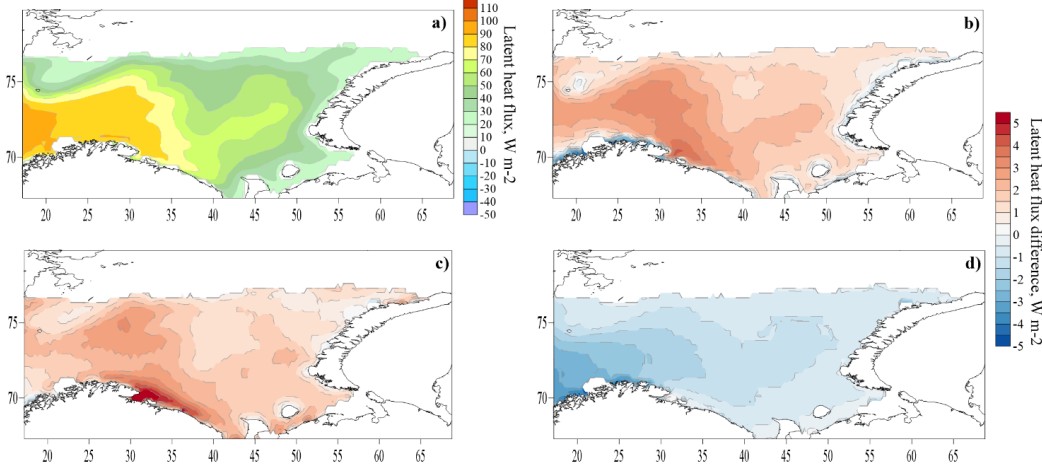


Figure 14. Mean latent heat flux in experiment C55 (a) and the flux difference in experiments T1 - C55 (b),
O2 - C55 (c) and D3 - C55 (d) during storms.

### 3.6 Turbulent heat fluxes during the cold-air outbreaks


Here we consider turbulent heat fluxes during cold-air outbreaks identified in Section 3.2 (2326 days with
cold-air outbreaks for the period 1979-2017). The average values of the sensible heat flux increase, especially in the
northwestern part (2 times compared with the average), during cold-air outbreaks (Figure 15a). The spatial
distribution of the latent heat flux is almost the same with the average one, but the flux magnitude increases by 1.5
times (Figure16a).
Experiments T1 and O2 everywhere provide to increase the magnitudes of the sensible and latent heat fluxes
compared to C55 during cold-air outbreaks (Figure 15, 16). Explicit accounting for the storm wave events leads to an
increase in heat fluxes mainly in the northwest of the sea and near the ice edge. But the differences between the
experiments are still small - on average less than 4 W m$^{-2}$ for the sensible heat flux and less than 2.5 W m$^{-2}$ for the
latent heat flux, i.e. less than 3-4% of flux magnitudes (Figure 15, 16). At the same time, the extreme values of the
flux difference during cold-air outbreaks, as for storm waves, are several times smaller than when considering long-
term means.
The average values of the flux difference during cold-air outbreaks are smaller than during storms, but the
extreme values during cold-air outbreaks and during storms are close.

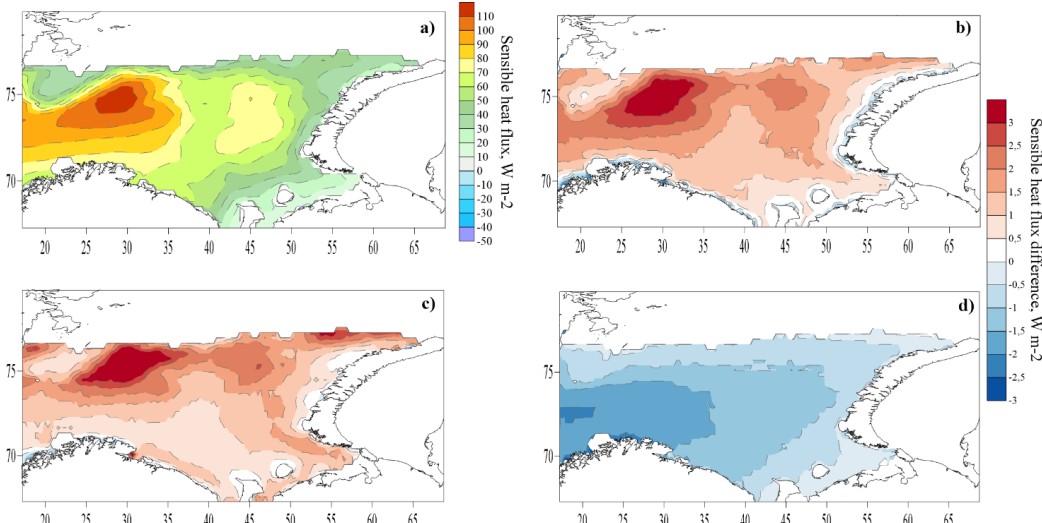


Figure 15. Mean sensible heat flux in experiment C55 (a) and the flux difference in experiments T1 - C55
(b), O2 - C55 (c) and D3 - C55 (d) during cold-air outbreaks.



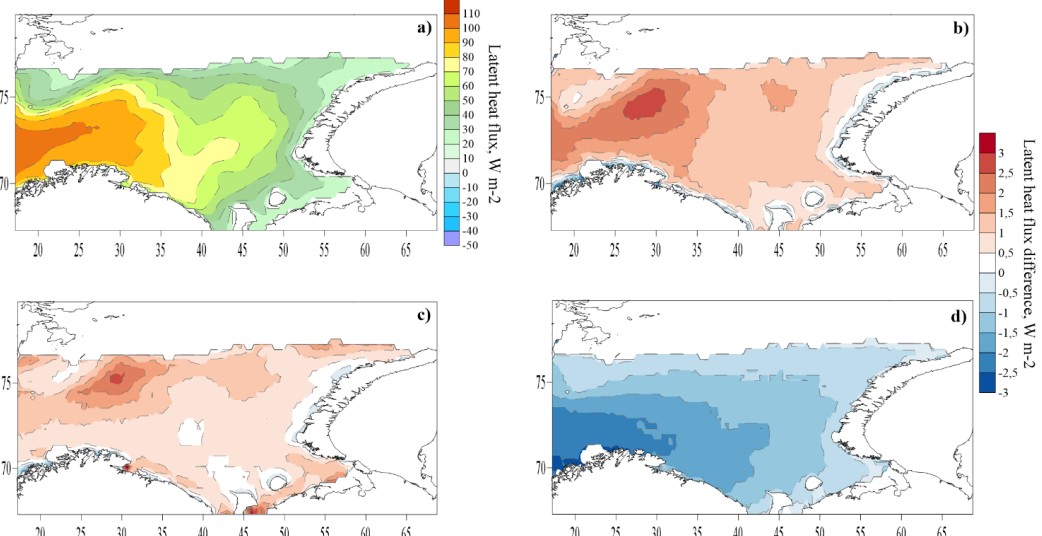


Figure 16. Mean latent heat flux in experiment C55 (a) and the flux difference in experiments T1 - C55 (b),
O2 - C55 (c) and D3 - C55 (d) during cold-air outbreaks.

**3.7 Turbulent heat fluxes during the simultaneously observed storm waves and cold-air outbreaks**
Finally, we consider cases when cold-air outbreaks and storm wave events were simultaneously observed (a
total of 292 days for the period 1979-2017) (Figure 17, 18). The magnitude of the heat fluxes and the difference
between the experiments in these cases are the largest in comparison with other situations. The sensible heat flux in
experiment C55 reaches 170 W m$^{-2}$ (in the north-west of the sea), the latent heat flux is 140 W m$^{-2}$ (in the west). The
average difference T1-C55 reaches 6 W m$^{-2}$ for sensible heat flux and 4.5 W m$^{-2}$ for latent heat flux. The average
difference O2-C55 reaches 10 W m$^{-2}$ for sensible heat flux and 7 W m$^{-2}$ for latent heat flux. The average difference
D3-C55 reaches 3 W m$^{-2}$ in the west of the sea.
The extreme values of the difference, which can reach 700 W m$^{-2}$, are also greatest in the case of
simultaneously observed storms and cold-air outbreaks. Figure 19 shows case when the difference in sensible heat
fluxes exceeded 100 W m$^{-2}$ between C55 and T1 parametrizations and 400 W m$^{-2}$ between C55 and O2
parametrizations. The greatest difference is noted for the eastern local maximum of the heat flux associated with the
cold-air outbreak during the north-eastern wind. An analysis of other cases, in which extreme values of the flux
difference were observed, also showed the presence of two local maxima (western and eastern) of heat fluxes. The
same maxima also appear in the long-term mean pattern of heat fluxes (Figure 15, 16).



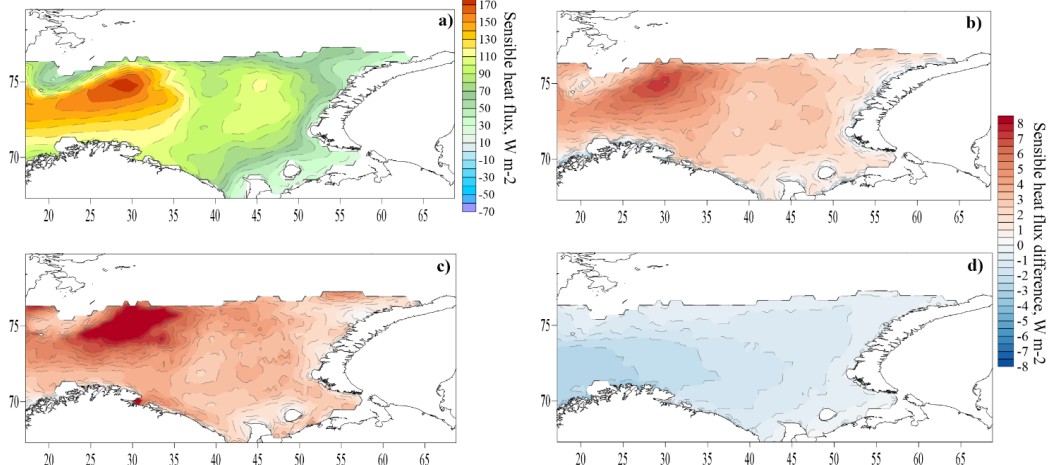

Figure 17. Mean sensible heat flux in experiment C55 (a) and the flux difference in experiments T1 - C55 (b), O2 - C55 (c) and D3 - C55 (d) during storms and cold-air outbreaks.

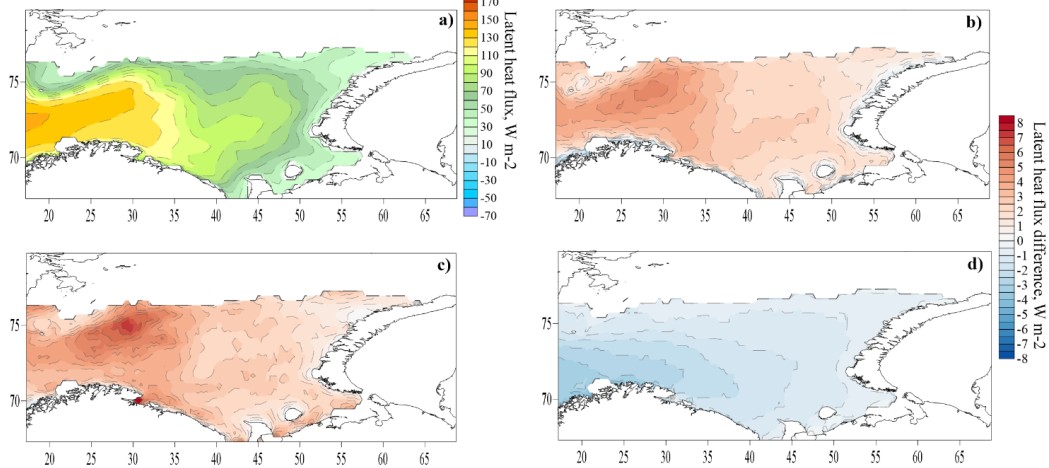

Figure 18. Mean latent heat flux in experiment C55 (a) and the flux difference in experiments T1 - C55 (b), O2 - C55 (c) and D3 - C55 (d) during storm waves and cold-air outbreaks.



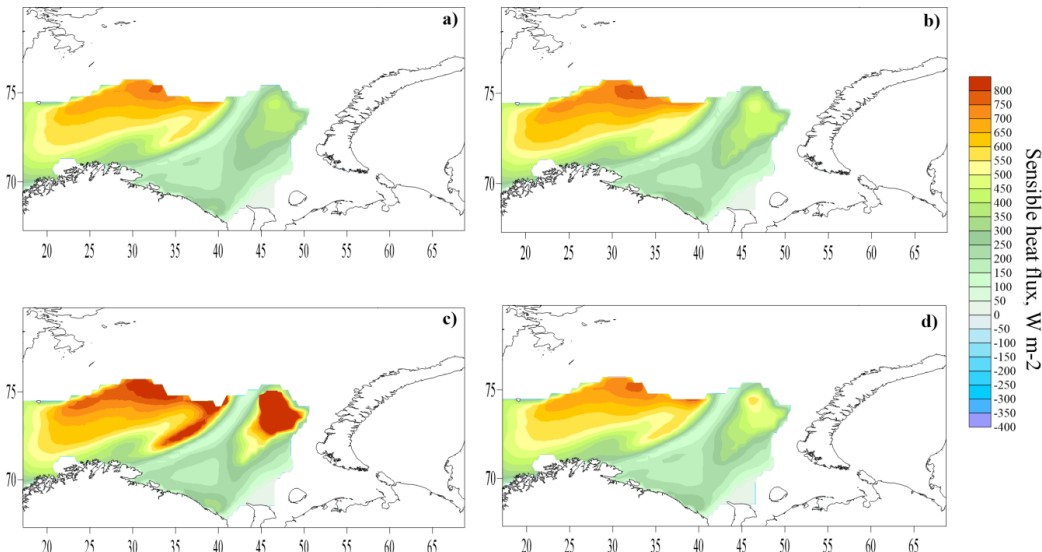

521

Figure 19. Sensible heat fluxes at 00 UTC January 13, 2003 calculated with C55 (a), T1(b), O2(c) and D3(d)

parametrizations.

### 4. Discussion and conclusions

This paper presents the results of turbulent heat flux calculations in the Barents Sea using the COARE

algorithm, meteorological data from reanalysis and sea-wave data from retrospective simulations with the WW3 wave

model. The calculations were performed for several options: using the modified Charnock parameterization of

roughness length (C55) and using the explicit accounting for the sea waves parameters in the roughness

parametrizations T1 (Taylor and Yelland), O2 (Oost et al.) and D3 (Drennan et al.). Particular attention was paid to

the episodes with extremely intense energy exchange between the atmosphere and the ocean: storms and cold-air

outbreaks (CAOs).

We obtained the mean annual distribution of the height and wavelength in the Barents Sea from wave

modelling results. Estimates of the storm activity from 1979 to 2017 were also obtained, confirming its high

interannual variability. Based on the data of wave modeling, a catalog of storm waves with the wave height exceeding

5 m was created. This catalog was used to calculate heat fluxes during storms.

The catalog of extreme CAOs over the Barents Sea was also obtained. It is shown that the extreme CAOs are

observed in 16.4% of days of a cold season (November-April). However, the number of CAO days varies from 6 in

2011-2012 to 56 in 1981-1982 manifesting large interannual variability. The important role of CAOs in the energy

exchange of the Barents Sea and the atmosphere is demonstrated. A high correlation was found between the number

of CAO days and turbulent fluxes of sensible and latent heat, as well as with the net flux of long-wave radiation

averaged over the ice-free surface of the Barents Sea during a cold season. Thus, the significant interannual variability

of the frequency of occurrence of CAOs largely determines the interannual variability of heat loss from the ice-free

surface of the Barents Sea.

Comparison of the calculated heat fluxes with ship observations during the NABOS expeditions was carried

out. Significant part of the errors in determining the heat fluxes is associated not with the used COARE algorithm, but

with discrepancies in meteorological parameters reproduced by the CFSR reanalysis and locally observed on the ship.



Excluding samples with big errors in meteorological parameters, we obtain the algorithm error of about 5-10 W m$^{-2}$,
which is within the accuracy of the eddy-covariance method during ship measurements.
The differences between the experiments (long-term calculations for the period 1979-2017) with different
parameterizations of the roughness length are small and are on average 1-3% of the flux magnitude. In some cases,
differences can reach 100-200 W m$^{-2}$. Parameterizations of Taylor and Yelland (2001) and Oost et al. (2002), which
represent the dependence of the roughness length on wave steepness and wave length, respectively, on average
overestimate the magnitude of the fluxes, and the parameterization of Drennan et al. (2003) (the dependence of the
roughness length on wave height and wave age) steadily underestimates the magnitude of the fluxes over the entire
sea compared to the Charnock parameterization. Thus, the effect of explicit accounting for wave parameters is, on
average, small and multidirectional, depending on the used parameterization. The modified Charnock formula quite
successfully describes the real behavior of the surface roughness even without explicitly taking into account the waves
parameters. This can be explained, firstly, by the Charnock parameter dependence on various ranges of wind speed
obtained from empirical data, and secondly, by the high correlation between wave parameters and wind speed usually
observed. In the climatic aspect, it can be argued that explicit accounting for sea waves in the calculations of heat
fluxes can be neglected.
However, in some situations, the choice of a particular roughness parameterization may be important. During
storms and cold-air outbreaks, differences between parameterizations increases along with the turbulent heat transfer
increase. In some extreme cases, during storms and cold-air outbreaks, the difference T1-C55 reaches 100 W m$^{-2}$, the
difference O2-C55 reaches 700 W m$^{-2}$.
The difference between the experiments with parameterization D3 and C55 is almost the same in all cases
and always decreases (modulo) from west to east of the sea, actually resembling the mean distribution of wave height.
Experiments with parameterizations T1 and O2 deviate most strongly from the Charnok parametrization in those areas
and at those times when the absolute values of the fluxes are large. The greatest absolute difference between the
fluxes is obtained for the simultaneous action of storms and cold-air outbreaks in the northwest and northeast of the
sea, i.e. when the values of the fluxes are the greatest. The relative flux difference (the difference normalized to the
value of the flux) over the entire sea is greatest during storms (in some areas more than 5%) (Figure 20), but in some
areas (in the north, near the ice edge), the relative difference is higher at the simultaneous action of cold-air outbreaks
and storms. In all situations, the relative difference is large in the region of the Pechora Sea due to the low absolute
values of the fluxes. An area of low absolute and relative values of the flux difference is located to the north-east from
Bear Island.




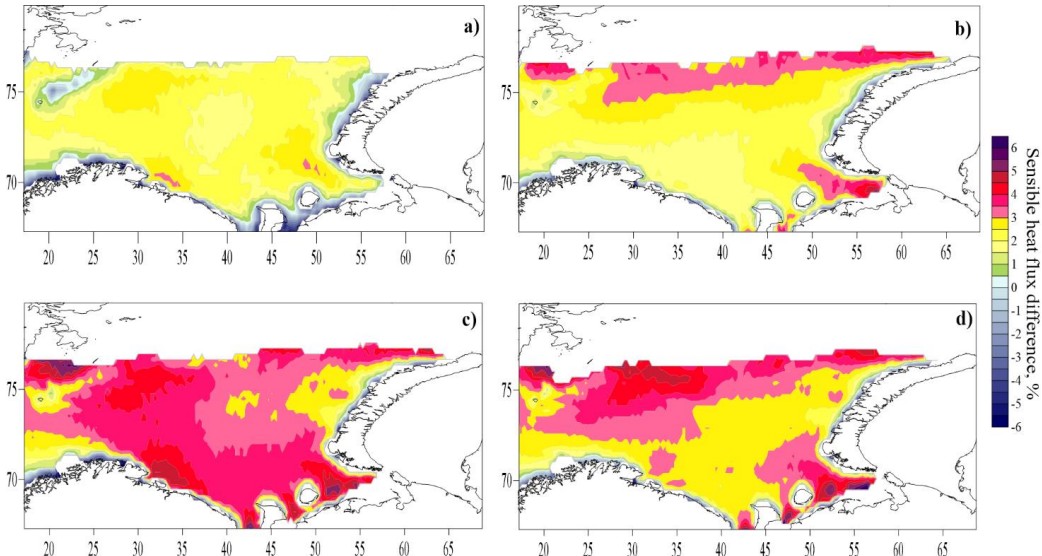

Figure 20. Mean relative difference in sensible heat flux (%) in the experiments T1 - C55 for all cases (a), during cold-air outbreaks (b), during storms (c) and during simultaneously observed storm waves and cold-air outbreaks (d).

Finally, based on the results of our study we can recommend the use of the parametrizations that take into account the wave parameters explicitly in the case of storms, cold-air outbreaks, and especially in the case of simultaneous action of storms and cold-air outbreaks, especially in the northern half of the sea, near the ice edge, in the Pechora Sea and near the coast of the Kola Peninsula (in case of storms).

**Data availability**

Data and results in this article resulting from numerical simulations are available upon request from the corresponding author.

**Author contributions**

The concept of the study was jointly developed by SM. SM did the numerical simulations, analysis, visualization and manuscript writing. ASh did the Coare simulations and its visualization. DCh did the calculations of cold-air outbreaks repeatability.

**Competing interests.**

The authors declare that they have no conflict of interest.

**Acknowledgments.**

Data analysis funded by the RFBR (project 18-05-60083 Shestakova A.A. and Chechin D.G.). The wave modeling was done with the financial support of the RFBR (project 20-35-70039 Myslenkov S.A.). Authors gratefully thank I.A. Repina for the provided shipborne observations collected during NABOS expeditions.

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
