# Peer review of "Sea waves impact on turbulent heat fluxes in the Barents"

_Atmospheric Chemistry and Physics, 2020_

## Referee Comment (RC1) · Anonymous Referee #1 · 24 Nov 2020

Comments on the manuscript "Sea waves impact on turbulent heat fluxes in the Barents 1 Sea according to numerical modeling" by Myslenkov et al.

General comments: This paper presents calculations of turbulent heat flux over the Barents Sea using the COARE algorithm, meteorological data from reanalysis and sea-wave data from wave model WWIII. Different parameterization schemes used to calculate heat fluxes Charnock (C55), Taylor and Yelland (T1), Oost et al. (O2) and Drennan et al. (D3). Spatial variations of latent and sensible heat fluxes as calculated from these methods are inter-compared for periods of storms and cold-air outbreaks (CAO) in the Barents Sea during winter season (November-April) of 1979-2017. The

ship-based observations (eddy-covariance method) are used to assess the derived heat fluxes. Based on the high correlation between the number of CAO days and turbulent fluxes of sensible and latent heat, it is concluded that the interannual variability of the frequency of occurrence of CAOs largely determines the interannual variability of heat loss from the ice-free surface of the Barents Sea in winter season. The differences in heat fluxes calculated from different parameterizations found to be small on average (1-3 % of the magnitude of flux). It also highlights errors in meteorological parameters in the reanalysis data which results into errors in calculation of turbulent heat fluxes.

The paper provides a comprehensive analysis of wave climate and spatial variations of latent and sensible heat fluxes in the winter-time ice-free regions of the Barents Sea. Further, the differences in heat fluxes (spatial maps in Barents Sea) during normal days, storm days, and CAO days calculated from different parameterization experiments are provided. In general, the paper is well-written. It contributes new knowledge to the field. The paper needs a revision in view of the following comments.

Specific comments: Based on the small differences in latent heat fluxes calculated from different parameterization experiments (Fig 9), it is concluded that an explicit accounting for sea waves in the calculations of heat fluxes can be neglected in climatic aspect. What would be the effect of background surface temperature and humidity conditions on the calculations of heat fluxes? How the differences in heat fluxes would look if calculated in mid-latitude or say in the tropical ocean? The effect of waves on heat fluxes can be region dependent (coastal to open ocean as seen in Figs 11, 13, 14), latitudinal differences, and wind conditions (very strong winds such as in hurricane can show larger heat flux difference from these parameterizations).

How good are the WWIII simulated wave parameters? Are there any in-situ measurements of wave parameters against which model results can be compared? How about wave model errors that might contribute to errors in heat flux calculations from equations 4, 5, and 6?

The number of storms per year plotted for significant wave heights (Fig 5) show a decline after 1995. Since the analysis is performed over a long time period of 1979-2017, how the sea ice changes (due to global warming) could impact the significant wave height, wave lengths? The CAO days (in Fig 6) also show somewhat declining trend.

Lines 388-402: It is interesting to note that the errors in calculation of heat fluxes dropped by more than 50% when the errors in reanalysis data (wind, temperature, humidity) are excluded. It points to the need of corrected reanalysis data product for a better estimate of heat fluxes. Enhanced in-situ measurements can help reanalysis data sets to overcome these bias. Surprisingly, the sensible and latent heat fluxes from different parameterizations are almost identical, even in high wind speed (or high Hs) cases.

Section 3.3: The ship based observations must be along the cruise-track of ship. Mention how the reanalysis input data for different parameterization methods/experiments is extracted for a comparison with ship data. Any area averaging was considered?

Figures 3 and 4: Are the significant wave heights and spectral peak wavelengths are shown for winter season or complete year during 1979-2017? Use color bar unit as 'm' in place of 'M'.

Figure 9: Time series of heat fluxes and significant wave heights are shown here. But, these measurements are not really continuous in time throughout. I suggest to have a break/gap in the continuous line joining data points when you jump from year 2005-2007 (2nd to 3rd data point), 2007 to 2013, 2013 to 2015.

Technical corrections: Line 16: '. . . .in the energy exchange of the Barents Sea and the atmosphere is. . . . . .' to be replaced with '. . . . . . ..in the energy exchange between the Barents Sea and atmosphere is. . . . . .'.

Line 73: Delete word 'also'.

Lines 77-80: 'The turbulent heat transfer . . . . . . . . . . . . . . . . . . . Brunke et al. 2011)'. Lengthy sentence difficult to read. Split the sentence into two.

Lines 80-81: '. . . . . .used in the main reanalyses. . . .' – Replace with '. . . . . . .used in different reanalyses. . . .'

Lines 96-98: Rephrase the sentence 'According to studies . . . . . . . . . . ..Barents Sea.' as 'According to studies of the wave climate of the Barents Sea (Wind and Wave. . ., 2003; Stopa et al., 2016; Liu Q. et al., 2016), stormy weather prevails during significant part of the year'. Also, check and correct the reference 'Wind and Wave. . ., 2003;' if mis-spelled.

Line 147: To be consistent with the acronym use 'WWIII' in place of 'WW3'.

Line 160: Add a reference to the Discrete Interaction Approximation (DIA) model or studies using it.

Define w' in equation 2.

Line 331: Do authors mean 'criteria (7m)' instead of criteria (7)?

Lines 409-410: 'The mean . . . . . . . . . . . . . . . . . on Figure 10, 11'. Check grammar and correct the sentence.

Line 423: Correct typo '-3 Ãů 2'.

---

## Referee Comment (RC2) · Anonymous Referee #2 · 3 Dec 2020

The paper describes the impact of the ocean surface waves in the calculation of the heat fluxes over the Barents Sea. Notably it aims to investigate the impact over the long term especially during storm waves events and cold-air outbreaks. A specific methodology is design to identify those events over the period of interest, 1979-2017. The Authors analyzed first the storm activity in the region and showed the correlation between the numbers of cold-air outbreaks days and the increase of heat fluxes. Then they studied the impact of waves on those heat fluxes by doing a comparison between several fluxes' parametrization using the COARE algorithm. They compared their results against ship measurements obtained during the NABOS campaigns over different years. The showed that over the long term the impact of waves on heat fluxes appears

to be small in average (1-3%). However, the difference during one single event could be significant. They conclude on the fact that in a climatic aspect taking into account waves in the parametrization of heat fluxes could be neglected.

Overall, the paper is generally written to a good standard, is relevant and has scientific merit. The methodology is well explained and the numerical set of simulation and experiments is appropriate to answer the scientific questions. The Authors bring valuable content for the scientific community and for the understanding of the air-sea processes over the Barents Sea region. The Authors should emphasis in their conclusion that the results are especially true in this region, according to their study. It might vary depending on the region and/or atmospheric conditions during other extreme events. I am content that the paper should be published following minor revision detailed below.

Wave modeling:

-What are the waves boundaries conditions used for the wave simulation? And so, as a related question how good does your wave simulation performed against observations and then if you have a bias could it also be a source of error when comparing the output of COARE against observations?

COARE input:

-If I understood correctly the wind seen by the waves in the simulation is the same as the one input in COARE, could you confirm that? Because if not it might include some inconsistency between the wind and waves parameters.

Heat fluxes difference:

-I found it interesting that on a single case you could have such a significant difference in heat fluxes between parametrizations, over 700 W/m2 which is quite large compared to the maximum value. And especially between parametrization which include waves. However, I would have liked a bit more explanation about this and what is causing this difference in the parametrization in your case, i.e. particular wind regime or sea state. I

think it would be worthy to explaine a bit further those differences since you mentioned them.

-Also, you recommend in the conclusion that it is better to use parametrization including wave parameters and I tend to be agreed with that. However, it is not clear for me according to your study because in one hand you showed quite small differences over the long-term average even during storm waves and CAO and on the other hand you showed large differences between parametrization over a case by case analysis. Can you be more precise on which parametrization we should use, or not use, especially for the Barents Sea region and/or during storm waves/CAO? Did you compare the parametrization against observations on a case by case analysis, for example the event showed on Fig 19?

Surface stress differences:

Did you look at the momentum fluxes differences between the parametrization in the long term and for CAO and storm waves? Since those are occurring during strong wind regime one could expect impacts on the roughness length and so the drag coefficient and surface stress. It might be worth mentioning it in the discussion or perspective since the stress is also an important factor in the air-sea processes and of COARE calculation.

Technical corrections:

Line 147 : "a development"

Line 315 : do you mean " and their detailed analysis would require an additional research" ?

Line 354 : ' is significant and represents up to 16%'

Fig 9: it would be better to show that it is a discontinuous data, that the gap between observations can be easily seen on the figure.

[Figure]

Line 411: "and difference between experiments are shown on"

Fig 10,11 : Do you mean " sea ice represents more than half of the grid nodes" ?

Line 424: fix typo 'within -3 $\sim$ 2'

Line 483:" Experiments T1 and O2 increase everywhere the magnitude of"

---

## Author Comment (AC1) · 3 Jan 2021

The authors are grateful for your comments. Specific comments: "Based on the small differences....Âż The conclusion that the impact of waves is small applies exclusively to the Barents Sea. This conclusion based on long-term calculations. Indeed, there are differences inside the Barents Sea, in some areas there is no influence, but somewhere more, but the maximum of 3%. Of course, heat fluxes depend on different parameters. The differences will be even smaller in the tropic or the equator regions, since there is a low storm activity. In the middle latitude (especially the southern hemisphere) the influence of waves on the heat fluxes probably will be more. However, we need to make a

long-term calculations to show the influence of humidity or temperature in similar wave conditions. We will add a comments on this topic. ÂńHow good are the WWIII...Âż The quality of our implementation corresponds to similar implementations of other authors. Correlation between model results and measurements data is 0.8–0.9, and the RMSE error is $\sim$ 0.3 m. There is very little direct measurement data. We can compare model results with satellite data to show errors. Since the scatter index of our version is 20-25%, then probably this value can lead to 2-3% errors in the final result in heat flux differences. ÂńThe number of storms per year...Âż Storms in the Barents Sea primarily come from the west with Atlantic cyclones. On this west side, the Sea is always open from ice. Reduction of ice slightly increases the number of storms which come from the north when the fetch in growing, but this is not visible in the long-term storm variability. Storms in the Barents Sea are more related to the Arctic Oscillation index. Cao events, on the contrary, are observed in the opposite atmospheric pressure situation – blocking of west-east transport. In theory, these graphs should not coincide. We will add comments on this topic. ÂńLines 388-402: It is interesting to note that the errors in calculation of heat fluxes dropped by more than 50% when the errors in reanalysis data (wind, temperature, humidity) are excluded. It points to the need of corrected reanalysis data product for a better estimate of heat fluxes. Enhanced in-situ measurements can help reanalysis data sets to overcome these bias. Surprisingly, the sensible and latent heat fluxes from different parameterizations are almost identical, even in high wind speed (or high Hs) cases.Âż Unfortunately, reanalysis errors are inevitable, especially in the Arctic, where there is little observational data to assimilate. However, we hope that these errors annihilate with a large time averaging. Small differences between parametrizations are explained by the prevalence of the developed sea state conditions, when all parametrization should behave well. For cases with young sea state difference in heat fluxes between parametrization reached 11% of the flux magnitude. Discussion of small differences between parametrizations will be added.

ÂńSection 3.3: The ship based observations must be along the cruise-track of ship. Mention how the reanalysis input data for different parameterization methods/experiments is extracted for a comparison with ship data. Any area averaging was considered?Âż CFSR and wave reanalysis were bilinearly interpolated (using 4 surrounding points) to ship location on every time step. No averaging was performed, since the reanalysis already has a rather coarse resolution and the values in its cells seem to correspond to the average value over the cell area. Figures 3 and 4: Yes, it is a Long-term average for complete year. "M" will be changed on "m".

ÂńFigure 9: Time series of heat fluxes and significant wave heights are shown here. But, these measurements are not really continuous in time throughout. I suggest to have a break/gap in the continuous line joining data points when you jump from year 2005- 2007 (2nd to 3rd data point), 2007 to 2013, 2013 to 2015.Âż We understand the remark, we will correct it.

All technical notes will be corrected. Thanks for your work!

---

## Author Comment (AC2) · 3 Jan 2021

The authors are grateful for your comments. "Wave modeling:. . ." We use calm conditions on the open boundaries. But we use a big unstructured grid which include all North Atlantic Ocean (waves do not come from south hemisphere to the Barents Sea, it is very far).On the North the ice fields is a nature boundaries. Therefore, our grid allows us to get correct estimates of wave climate in the Barents Sea.

" COARE input: If I understood correctly . . ...." Yes, the wind input in the wave model and in COARE is the same: it is 10-m wind from CFSR reanalysis. "Heat fluxes difference: I found it interesting . . .." Some analysis of these significant differences will be

added to the text. Maximal differences between parametrizations are observed for the young sea state.

"Also, you recommend in the conclusion . . .." The choice on neglecting or not neglecting the explicit wave account depends on the application. In climate studies operating with large time-scales and spatially and temporally averaged values (for example, in future climate modeling) the difference between parametrizations is small and the Charnock parametrization (which do not involve additional wave modelling) seems to be sufficient. On smaller time scales, for example, in weather prediction, the choice of parametrization plays a greater role. However, it is impossible to determine the best parametrization because there are no in-situ measurements of heat fluxes in those areas and those times, where heat flux differences in parametrizations are big. Available measurements shown on Fig.9 corresponds to situations when differences between parametrizations were rather small. Some explanations on this topic will be added to the text.

"Did you look at the momentum fluxes differences . . . . . .." In this paper, we focused specifically on heat fluxes, since Barents Sea is a "hot spot" in terms of heat air-sea exchange. However, we will add a few sentences on momentum flux at the end of the paper.

Technical corrections:

Line 147 : "a development" - OK.

Line 315 : do you mean " and their detailed analysis would require an additional research" ? - YES

Line 354 : ' is significant and represents up to 16%' - OK

Fig 9: it would be better to show that it is a discontinuous data, that the gap between observations can be easily seen on the figure. -OK

Line 411: "and difference between experiments are shown on" -OK
Fig 10,11 : Do you mean " sea ice represents more than half of the grid nodes" ? No. It mean that the sea ice in one node was in 50% of all time of calculations.

Line 424: fix typo 'within -3 _ 2' – OK. Line 483:" Experiments T1 and O2 increase everywhere the magnitude of" – OK.

Thanks for your work!

---

## Author Response (AR1)

**Reviewer 1**

The authors are grateful for your comments.

Specific comments:

*"Based on the small differences in latent heat fluxes calculated from different parameterization experiments (Fig 9), it is concluded that an explicit accounting for sea waves in the calculations of heat fluxes can be neglected in climatic aspect. What would be the effect of background surface temperature and humidity conditions on the calculations of heat fluxes? How the differences in heat fluxes would look if calculated in mid-latitude or say in the tropical ocean? The effect of waves on heat fluxes can be region dependent (coastal to open ocean as seen in Figs 11, 13, 14), latitudinal differences, and wind conditions (very strong winds such as in hurricane can show larger heat flux difference from these parameterizations)»*

The conclusion about the small impact on waves applies exclusively to the Barents Sea. This conclusion based on long-term calculations. Indeed, there are differences inside the Barents Sea, in some areas there is no influence, but somewhere more, but the maximum of 3%.

Of course, heat fluxes depend on different parameters. The differences will be even smaller in the tropic or the equator regions, since there is a low storm activity. In the middle latitude (especially the southern hemisphere) the influence of waves on the heat fluxes probably will be more. However, we need to make a long-term calculations to show the influence of humidity or temperature in similar wave conditions. We add a comments on this topic at lines 600-602

*«How good are the WWIII simulated wave parameters? Are there any in-situ measurements of wave parameters against which model results can be compared? How about wave model errors that might contribute to errors in heat flux calculations from equations 4, 5, and 6?»*

The quality of our implementation corresponds to similar implementations of other authors. Correlation between model results and measurements data is 0.8–0.9, and the RMSE error is ~ 0.5 m. We compared wave model results with satellite data to show model quality. We add a Fig.2 and comments about wave modeling quality at lines 190-194.

Since the scatter index of our modeled significant wave heights is 0.28 (or 28%), then probably this value can lead to mean errors ~4-5% in the calculated heat flux values when the wave heights is ~ 5 m. We add this comments at lines 249-250.

*«The number of storms per year plotted for significant wave heights (Fig 5) show a decline after 1995. Since the analysis is performed over a long time period of 1979-2017, how the sea ice changes (due to global warming) could impact the significant wave height, wave lengths? The CAO days (in Fig 6) also show somewhat declining trend»*

Storms in the Barents Sea primarily come from the west with Atlantic cyclones. The Barents Sea is always open from the west side from ice. Reduction of ice slightly increases the number of storms which come from the north when the fetch in growing, but this is not visible in the long-term storm variability. Storms in the Barents Sea are more related to the Arctic Oscillation index. Cao events, on the contrary, are observed in the opposite atmospheric pressure situation – blocking of west-east transport. In theory, these graphs should not coincide. We add comments on this topic at lines 359-365 and 396-401.

*«Lines 388-402: It is interesting to note that the errors in calculation of heat fluxes dropped by more than 50% when the errors in reanalysis data (wind, temperature, humidity) are excluded. It points to the need of corrected reanalysis data product for a better estimate of heat fluxes. Enhanced in-situ measurements can help reanalysis data sets to overcome these bias. Surprisingly, the sensible and latent heat fluxes from different parameterizations are almost identical, even in high wind speed (or high Hs) cases.»*

Unfortunately, reanalysis errors are inevitable, especially in the Arctic, where there is little observational data to assimilate. However, we hope that these errors annihilate with a large time averaging. Small differences between parametrizations are explained by the prevalence of the developed sea state conditions, when all parametrization should behave well. For cases with young sea state difference in heat fluxes between parametrization reached 11% of the flux magnitude. Discussion of small differences between parametrizations we add in section "Discussion…"

*«Section 3.3: The ship based observations must be along the cruise-track of ship. Mention how the reanalysis input data for different parameterization methods/experiments is extracted for a comparison with ship data. Any area averaging was considered?»*

CFSR and wave reanalysis were bilinearly interpolated (using 4 surrounding points) to ship location on every time step. No averaging was performed, since the reanalysis already has a rather coarse resolution and the values in its cells seem to correspond to the average value over the cell area.

*"Figures 3 and 4:"*

Yes, it is a long-term average for complete year. "M" changed on "m".

*«Figure 9: Time series of heat fluxes and significant wave heights are shown here. But, these measurements are not really continuous in time throughout. I suggest to have a break/gap in the continuous line joining data points when you jump from year 2005- 2007 (2nd to 3rd data point), 2007 to 2013, 2013 to 2015.»*

We understand the remark, figure corrected.

*"Line 331: Do authors mean 'criteria (7m)' instead of criteria (7)?"*

It mean formula (7), we add clarification.

All technical corrections fixed. Thanks for your work!

**Reviewer 2**

The authors are grateful for your comments.

*"What are the waves boundaries conditions used for the wave simulation? And so, as a related question how good does your wave simulation performed against observations and then if you have a bias could it also be a source of error when comparing the output of COARE against observations?"*

We use calm conditions on the open boundaries. However, we use a very big unstructured grid which include all North Atlantic Ocean (waves do not come from south hemisphere to the Barents Sea, it is very far). On the North boundary the ice fields is a nature boundaries. Therefore, our grid allows us to get correct estimates of wave climate in the Barents Sea.

We compared wave model results with satellite data to show model quality. We add a Fig.2 and comments about wave modeling quality at lines 190-194.

Since the scatter index of our modeled significant wave heights is 0.28 (or 28%), then probably this value can lead to mean errors ~4-5% in the calculated heat flux values when the wave heights is ~ 5 m. Thus, the differences between the output of COARE and observations in not caused by errors in wave modeling.

*« If I understood correctly the wind seen by the waves in the simulation is the same as the one input in COARE, could you confirm that? Because if not it might include some inconsistency between the wind and waves parameters...»*

Yes, the wind input in the wave model and in COARE is the same: it is 10-m wind from CFSR reanalysis.

*« Heat fluxes difference: I found it interesting ….»*

Some analysis of these significant differences we add to the text at lines 539-550. Maximal differences between parametrizations are observed for the young sea state.

*«Also, you recommend in the conclusion that it is better to use parametrization including wave parameters and I tend to be agreed with that...»*

The choice on neglecting or not neglecting the explicit wave account depends on the application. In climate studies operating with large time-scales and spatially and temporally averaged values (for example, in future climate modeling) the difference between parametrizations is small and the Charnock parametrization (which do not involve additional wave modelling) seems to be sufficient. On smaller time scales, for example, in weather prediction, the choice of parametrization plays a greater role. However, it is impossible to determine the best parametrization because there are no in-situ measurements of heat fluxes in those areas and those times, where heat flux differences in parametrizations are big. Available measurements shown on Fig.9 corresponds to situations when differences between parametrizations were rather small. Some explanations on this topic we add at lines 628-640.

*«Did you look at the momentum fluxes differences between the parametrization in the long term and for CAO and storm waves? Since those are occurring during strong wind regime one could expect impacts on the roughness length and so the drag coefficient and surface stress. It might be*

*worth mentioning it in the discussion or perspective since the stress is also an important factor in the air-sea processes and of COARE calculation.»*

In this paper, we focused specifically on heat fluxes, since Barents Sea is a "hot spot" in terms of heat air-sea exchange. However, we add a few sentences on momentum flux at the "discussion section" of the paper.

*Fig 10,11 : Do you mean " sea ice represents more than half of the grid nodes" ?*

No. It mean that the sea ice in one node was in 50% of all time of calculations.

*Other Technical corrections fixed.*

Thanks for your work!